# Reconciling grain growth and shear-coupled grain boundary migration

Spencer L. Thomas[1], Kongtao Chen[1], Jian Han[1], Prashant K. Purohit[2] & David J. Srolovitz [1,2]

Conventional models for grain growth are based on the assumption that grain boundary (GB) velocity is proportional to GB mean curvature. We demonstrate via a series of molecular dynamics (MD) simulations that such a model is inadequate and that many physical phenomena occur during grain boundary migration for which this simple model is silent. We present a series of MD simulations designed to unravel GB migration phenomena and set it in a GB migration context that accounts for competing migration mechanisms, elasticity, temperature, and grain boundary crystallography. The resultant formulation is quantitative and validated through a series of atomistic simulations. The implications of this model for microstructural evolution is described. We show that consideration of GB migration mechanisms invites considerable complexity even under ideal conditions. However, that complexity also grants these systems enormous flexibility, and that flexibility is key to the decades-long success of conventional grain growth theories.

[1] Department of Materials Science and Engineering, University of Pennsylvania, Philadelphia, PA 19104, USA. [2] Department of Mechanical Engineering and Applied Mechanics, University of Pennsylvania, Philadelphia, PA 19104, USA. Correspondence and requests for materials should be addressed to D.J.S. (email: srol@seas.upenn.edu)

The conventional theory of capillarity-driven grain growth is simply grain boundary (GB) curvature flow[1], in which each GB segment migrates toward its center of mean curvature with a velocity

$$v = M\gamma\kappa, \qquad (1)$$

where $M$ is a temperature-dependent mobility, $\gamma$ is the GB energy, and $\kappa$ is the GB mean curvature. In the general case, the prefactor of the curvature depends on the five macroscopic properties that describe GB bicrystallography. However, even such generalized curvature flow grain growth models fail to explain many common observations from grain growth experiments and simulations, such as stress-assisted grain growth[2–5], grain rotation[4, 6, 7], GB sliding[2, 8, 9], and abnormal grain growth[10, 11].

Previously, variations in GB mobility has been proposed as a potential driver of abnormal grain growth[12] and grain rotation[13, 14]. Solutes/impurities provide a drag on GB migration[15] and may even stabilize nanocrystalline microstructures either by kinetic pinning[16] or thermodynamic stabilization[17], and may influence abnormal grain growth[16, 18]. However, it is our conjecture that many deviations from conventional curvature-driven boundary migration may be attributed to the intrinsic mechanisms by which GBs migrate.

Observations that shear stress can drive GB migration—an effect not addressed by conventional GB migration models—originated over a half-century ago[19, 20]. Conversely, in such shear-coupled migration[21], GB migration can induce the translation of one grain with respect to the other. Reversing the sign of shear reverses the direction of migration and vice-versa. Shear-coupled GB migration is generally characterized by the temperature-dependent ratio of the grain translation rate $\dot{B}$ and the GB migration rate $\dot{H}$

$$\beta = \frac{\dot{B}}{\dot{H}}. \qquad (2)$$

Observations of shear-coupling in high-angle GBs have been reported in experiments[22–24], *ab initio* calculations[25–27] and atomistic simulations[28–31]. Shear-coupling was identified as a general GB property[32] related to the dislocation content of GBs[33, 34]. While it may seem intuitively obvious for low-angle tilt GBs consisting of arrays of glissile edge dislocations, shear-coupled migration is less obvious for high-angle GBs. Cahn et al.[35] showed that total dislocation content of the GB[36–38] can be used to predict shear-coupling even for high-angle GBs, where

individual dislocations are not easily resolved. Recent theoretical predictions[39–42] and transmission electron microscopy (TEM) observations[43] connected shear-coupling directly to a disconnection mechanism of GB migration[44]. A disconnection is a line defect that lies within a GB and is characterized by a Burgers vector and step-height, the allowed values of which depend on the macroscopic geometric parameters describing the GB (grain misorientation and GB plane inclination). The shear-coupling parameter $\beta$ is related to the Burgers vector and step-height associated with the disconnections.

While most of the modeling and theory literature focuses on the behavior of individual, flat grain boundaries, the motion of GBs within a polycrystal is inevitably constrained by other grains and GB junctions. It is in such polycrystalline systems (where GB migration matters) that many unconventional effects are observed in experiment and simulation. For example, grain rotation[6, 7, 45] may be related to the misorientation-dependence of GB energy[13, 14, 46] or grain coalescence[47–50], but it may also be a natural consequence of shear-coupling[32]. Twinning, commonly observed in many close-packed materials, is a deformation mechanism that is often observed to occur during grain growth. Such deformation and annealing twinning may share a common mechanism if grain growth is accompanied by stress generation associated with shear-coupling. Finally, we note that the application of stress has been shown to accelerate grain growth[2]. This too is not described by curvature flow models, but may also be rationalized in terms of shear-coupling.

Here, we examine the effect that constraints present in all polycrystalline materials have on shear-coupled GB migration, and develop a predictive understanding of the interaction between shear-coupling and grain growth that reconciles the many unconventional grain growth phenomena outlined above. This new understanding carries deeper implications to grain growth theory; the disconnection model suggests that stress generation, grain rotation, defect formation, and abnormal grain growth should all be general features of grain growth, but it also introduces a degree of flexibility that facilitates more conventional grain growth behavior.

## Results

**Atomistic simulation of grain growth.** We first perform a series of MD simulations of grain growth in nanocrystalline Ni to identify essential features of microstructure evolution, see Fig. 1. Significant grain growth occurs during the 2.5 ns simulation; the

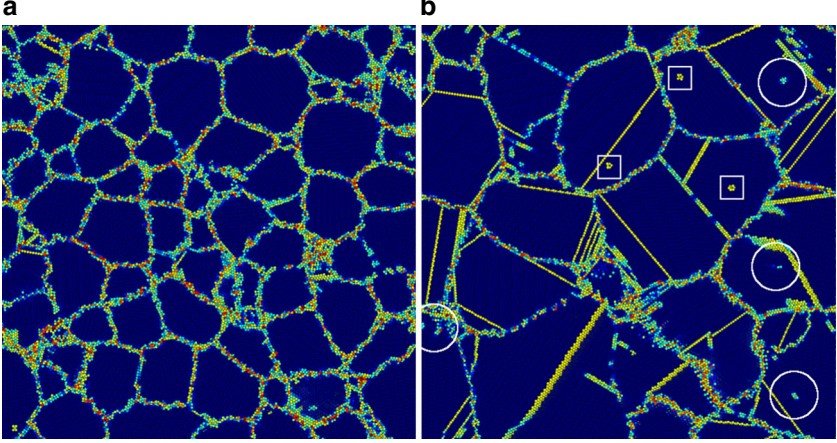

**Fig. 1** MD polycrystalline grain growth. Cross-section of a three-dimensional MD grain growth simulation cell. **a** Initial relaxation (see Methods). **b** The same cross-section following a 2.5 ns anneal at $0.85T_m$. Atom colors are assigned based on centrosymmetry. The white circles indicate dislocations, white squares identify vacancies, and thin yellow lines show twins formed during grain growth. For a full animation of this simulation, see Supplementary Video 1

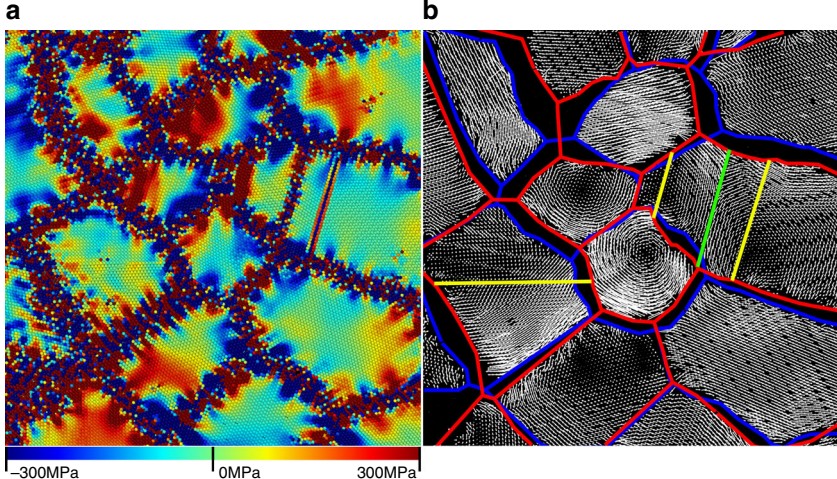

**Fig. 2** Stress generation and lattice rotation in MD polycrystalline grain growth. A small region of an *xy* cross-section of the simulation shown in Fig. 1 at 1.25 ns. **a** $\sigma_{xy}$ showing large accumulations of stress (dark blue and dark red regions) inside grains. Stresses along GBs reflect the local GB structure and are not pertinent to this discussion. **b** Displacement vectors depicting the motion of atoms between 1.0 and 1.25 ns after initial relaxation. Red and blue lines indicate GB positions at 1.0 and 1.25 ns. Yellow and green lines indicate coherent twin boundaries and stacking faults. Lattice rotations (cyclic displacement vector patterns) and translation (large regions of parallel displacement vectors) are both apparent. Atoms with very large displacements (such as where the GB migrated) were removed for clarity

mean linear grain size $\ell$ increases from 61 to 126 Å. A substantial number of defects, including vacancies, dislocations, and twins, also form during these simulations. Most of these defects form during grain growth rather than in the initial relaxation; this implies that defect generation is a consequence of GB migration in the polycrystal. Dislocation and twin formation and propagation are widely associated with large stresses and generally serve as a stress-relaxation mechanism. In particular, we observe the formation of sets of parallel twins in the wake of migrating triple junctions (TJs). TJs prevent shear at the ends of a migrating, shear-coupled GB, and hence are sites of severe stress concentration. Twinning near such migrating GBs may relax these stresses.

Direct measurements of stress (Fig. 2a) in the polycrystalline simulations reinforce this stress-generation argument. Some grains exhibit large internal stresses, while others show only small or zero stresses. As shear stresses within the grains are very small prior to grain growth, this stress generation must be associated with GB migration. GB migration is also accompanied by grain rotation (see Fig. 2b). While grain rotations are sometimes attributed to torque on the grains associated with the misorientation-dependence of GB energy[13, 46] or with grain coalescence[47–50], no grain coalescence or disappearance occurs in the vicinity of the rotating grains in Fig. 2 during the time-span in the displacement vector plots. Lattice rotation is ubiquitous in these simulations, implying that it is a general feature of grain growth.

The stress generation and lattice rotation observed in the grain growth simulation could be induced by shear-coupled GB migration. Most shear-coupling studies focus on flat GBs of infinite extent (or in periodic systems). However, GBs in a polycrystal are finite; each grain is delimited by multiple grains and each GB is delimited by GB triple junctions (TJs). Little is known about the effects of such constraints on the evolution of a polycrystal. Shear displacement across a GB plane will be limited by these TJs; a disconnection cannot propagate from one GB to another because each GB bicrystallography has a unique set of disconnections. This implies that shear-coupling along a GB of finite length necessarily generates stress at the TJs. How can a GB migrate large distances if it is shear-coupled and generates stress

at the TJs proportional to its migration distance? Clearly, GB migration during grain growth corresponds neither to conventional curvature-driven migration nor to ideal shear-coupling.

Polycrystalline MD simulations hint at what is missing in conventional grain growth models. However, such simulations are too complex for detailed analysis of what is occurring in every grain, every GB, and every TJ. Instead, we turn to a simpler, idealized microstructure that exhibits many features of polycrystalline grain growth, but is more amenable to detailed analysis.

**Idealized polycrystalline microstructure.** We construct a simple, idealized, three-dimensional (3D) microstructure with just a few grains and a small set of grain boundaries, as shown in Fig. 3a. The lattice orientation of each grain and the mobility (*M*) and shear-coupling factor (*β*) for each GB is given in Supplementary Tables 1 and 2. The temporal evolution of the idealized microstructure is shown in Fig. 3. The central four-sided grain (B in Fig. 3a) shrinks and disappears, while the outer square grains (A) do not. This is inconsistent with conventional grain growth theory, which implies that for such 2D microstructures, the area *A* of an *n*-sided grain will evolve according to the von Neumann–Mullins relation, $\frac{dA}{dt} = \frac{\pi}{3} M\gamma(n-6)$[51, 52]. While a three-dimensional version of the von Neumann–Mullins relation exists[53], this two-dimensional form applies here because the grains are columnar. All four-sided (*n* = 4) grains should shrink at the same rate, provided all of the surrounding GBs have identical *M*s and *γ*s. Supplementary Table 2 shows that this is true to within 15% for both Grains A and B. However, Grain B shrinks and disappears, while Grain A changes very little during the present simulation. Therefore, capillarity is an insufficient description of microstructure evolution, even in this simple case.

Most of the main features (stress generation, lattice rotation) that occur in the general microstructure (Fig. 1) are reproduced in the simple microstructure (Fig. 3) (no dislocations or twins are formed, but dislocation slip and twinning occur predominantly on {111} planes in FCC metals and the [111] direction is along the axes of each columnar grain, so this is unsurprising). Figure 4 shows that Grain A (which does not shrink) develops a large

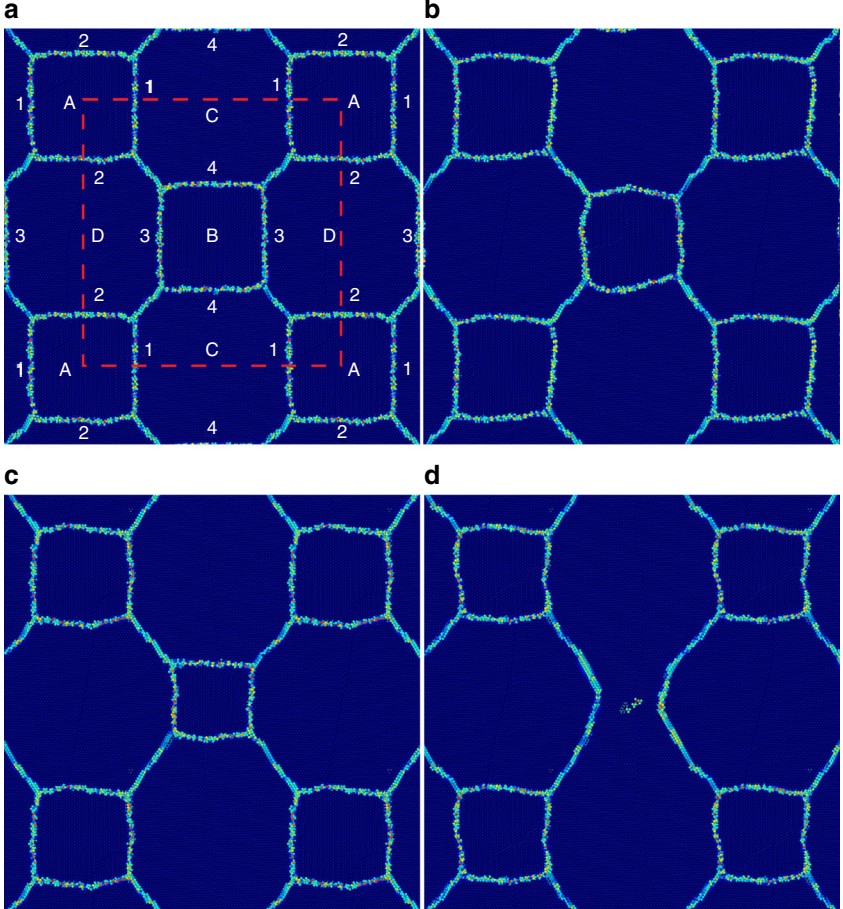

**Fig. 3** Idealized polycrystalline microstructure. Time Evolution of the columnar 3D microstructure. **a** Initial microstructure with grain and GB labels. The dashed red line indicates the periodic simulation cell. Crystallographic details are provided in Supplementary Table 1. **b–d** Evolution of the microstructure in **a** For a full animation of this simulation, see Supplementary Video 2

shear stress, but rotates little, while the stress in grain B (which does shrink) is small and the rotation is significant. Conventional grain growth theory cannot account for the observed lattice rotation, stress generation, or arrested GB migration, but shear-coupling may. (Note that while a misorientation-dependent grain boundary energy could produce a torque that may rotate a grain, this would require long-range material transport and would likely not be observable during the time-scale of the molecular dynamics simulation in Fig. 4). Lattice rotation may result from shear-coupling when all of the GBs bounding a grain have the same coupling sense (e.g., clockwise); this depends on the signs of the coupling parameters $\beta$ for each GB. Meanwhile, if the shear-coupled displacements are not of the same sense, no rotation will occur, and a shear stress must develop during GB migration. If this stress is large enough, it can produce an elastic driving force for GB migration in the direction opposite that of capillarity. Hence, GB migration will slow or stagnate. In the next section, we examine these observations through investigation of individual GBs.

**Shear-coupled bicrystal simulations**. Shear-coupled migration is illustrated in Fig. 5 for a Ni bicrystal that is periodic in the direction parallel to the symmetric tilt GB ($\Sigma 39[111]\theta = 32.2°$) and free at the top and bottom of the MD simulation cell. We drive GB migration via a difference in energy density between the two crystals (a synthetic driving force[54]), $\Psi$. As the GB migrates, it creates a shear displacement, which is visualized via the fiducial

line (a group of Ni atoms colored red). In Fig. 5, the slope of this line is the inverse of the shear-coupling parameter $\beta^{-1} = 1/0.58$.

Shear-coupling can be understood in terms of the nucleation and motion of disconnections along the GB[39–42]. The glide of a disconnection with Burgers vector $b$ (the component of $\boldsymbol{b}$ parallel to the GB plane) and step-height $h$ shifts the two crystals by $b$ parallel to the GB and displaces the GB (normal to itself) by $h$. Microscopically, the shear-coupling factor associated with disconnection $i$ is $\beta_i = b_i/h_i$. For any particular GB, $(\boldsymbol{b}, h)$ is not unique; there is a series of possible disconnections $\{(\boldsymbol{b}_i, h_j)\}$ for each GB determined by bicrystallography[55]. We distinguish between a macroscopic value of $\beta$, which is temperature-dependent, and reflects the observed shear-coupling behavior (i.e., $\beta = \dot{B}/\dot{H}$) and those associated with a particular disconnection $\{(\boldsymbol{b}_i, h_j)\}$, $\beta_i$. At low temperature, the expected value of $\beta$ corresponds to the disconnection mode $\{(\boldsymbol{b}_i, h_j)\}$ with the lowest nucleation barrier under the current driving force (denoted $i = 0$), $\beta = \beta_0$[39, 56].

Figure 6 depicts a simulation with exactly the same initial atomic configuration, temperature, driving force, and simulation dimensions as in Fig. 5, but for which the top and bottom ends of the simulation cell are held fixed (rather than free). Under these conditions, the GB migrates a short distance, then arrests. Figure 6d shows the GB position and shear stress vs. time for this simulation. As the upper and lower edges of the simulation cell cannot freely translate, a shear stress accumulates due to shear-coupled GB migration. This results in an elastic driving force that opposes migration. For an energy density difference between two

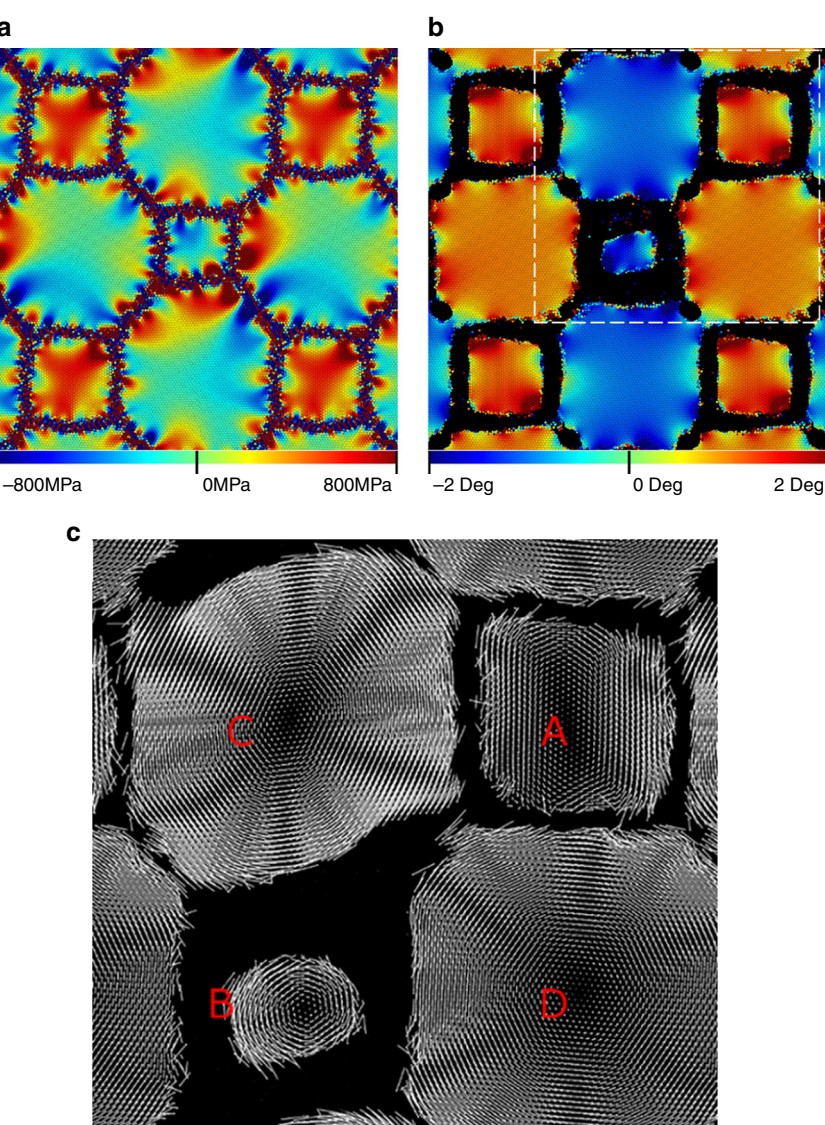

**Fig. 4** Stress generation and lattice rotation in the idealized microstructure. **a** $\sigma_{xy}$, **b** grain rotation, and **c** displacement vectors from 5.5 ns into the evolution of the idealized microstructure. There is a large positive stress accumulation in Grain A and lattice rotation in Grain B. Displacement vectors are scaled by a factor of 20 for visual clarity and grain labeling in **c** is consistent with Fig. 3a. The white box in **b** denotes the viewing boundaries in **c**. Displacement vectors for atoms with very large displacements (such as those through which a GB has migrated) were removed for clarity. The rotation in **b** was calculated as the curl of the displacement field depicted in **c**. For full animations corresponding to **a**, **b** see Supplementary Videos 3 and 4, respectively

grains $\Psi$, the total driving force tends to zero at a critical stress $\tau_c = -\Psi/\beta$ (this is quantitatively consistent with the simulation data in Fig. 6d). We return to this prediction below.

While these simulations focus on the migration of single, flat GBs, they emulate one of the constraints that occurs in real microstructures; one grain cannot freely translate with respect to the other because of the presence of surrounding grains. While the restriction in the case of the polycrystal is associated with the surrounding grains in the polycrystal, the fixed-end bicrystal simulations provide a simple analog. However, unlike in the fixed-end bicrystal simulations where GB migration stops, many GBs in polycrystals are able to migrate long distances. To examine this apparent contradiction, we consider the migration of another GB under similar constraints.

Figure 7 shows the migration of a $\Sigma13[111]\theta = 27.8°$ symmetric tilt GB under the same fixed-end constraints as in Fig. 6. The GB initially migrates with $\beta = 0.50$. However, instead of stagnating, this GB switches the coupling parameter sign (a change in the

sign of the fiducial line slope) to $\beta = -0.58$ and continues to migrate. It migrates with this new coupling sense for a finite distance, then switches back to the initial coupling parameter. This results in the zig-zag pattern in the fiducial mark in Fig. 7. Figure 7f shows that the stress initially builds as the GB migrates, then relaxes when $\beta$ switches signs. Rather than stagnating, the GB continues to migrate via this switch-back mechanism. Note that the average stress is non-zero during this "steady" migration.

Both grain boundary stagnation and disconnection mode-switching are possible during GB migration. In general, mode-switching is necessary to permit long-distance GB migration. Not all GBs migrate in the same manner, and even a single GB may not migrate in the same fashion under all conditions. Mode-switching depends not only on the accessibility of secondary disconnection modes (difference in disconnection formation energies for different modes), but also on the local microstructure. For example, the local microstructure determines the direction and degree to which the GBs surrounding a particular

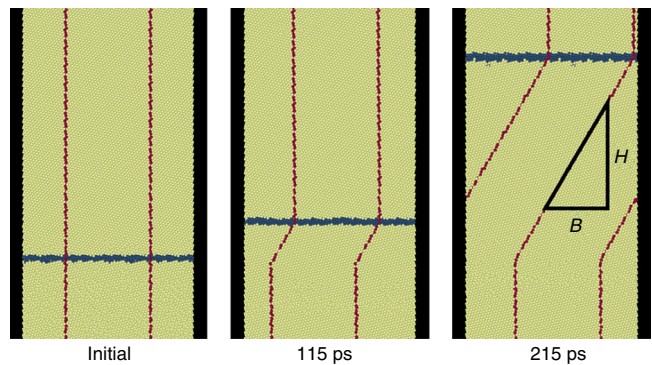

**Fig. 5** Shear-coupling in a symmetric tilt GB with free ends. Time sequence depicting shear-coupling in a $\Sigma 39[111]\theta = 32.2°$ symmetric-tilt GB for a difference in energy density between the two crystals[54] $\Psi = 8.0$ meV Å$^{-3}$ at 300 K with free top and bottom surfaces. The blue horizontal and red vertical lines are the GB position and fiducial mark (red atoms), respectively. As the GB migrates, the upper grain shifts right. Two periods of the simulation cell are shown in the horizontal direction and the free top and bottom surfaces are outside the sections of the simulation cell shown

grain shear-couple, what disconnection reactions are possible at GB triple junctions, and what stresses result at the GB from processes within the grain (e.g., plasticity).

**Single disconnection mode.** We construct a simple elastic model to describe grain boundary stagnation as observed in the constrained shear-coupling case shown in Fig. 6. In Fig. 8d, we consider the lateral displacement field $u_x(y, t)$ with respect to the reference configuration (Fig. 8a) for cases where the GB migrates from $H_0 \rightarrow H$ and a macroscopic displacement gradient $\tan \gamma$ is applied:

$$u_x(y, t) = \begin{cases} y \tan \gamma & y < H_0 \\ y \tan \gamma + (y - H_0)\beta & H_0 < y < H \\ y \tan \gamma + (H - H_0)\beta & y > H, \end{cases} \quad (3)$$

where $\beta = B/(H - H_0)$ (Fig. 8c). Since $\beta$ is constant, this definition is equivalent to $\dot{B}/\dot{H}$. The lateral displacement at the top of the cell (Fig. 8d) is

$$D(t) = u_x(L, t) = L\gamma + \beta(H - H_0), \quad (4)$$

where we have made the small strain approximation $\tan \gamma \approx \gamma$. For constant $\Psi$, it follows (see the Supplementary Note 4 for the detailed derivation) that

$$\dot{H} = M(\Psi + \beta\tau) \quad (5)$$

$$\dot{D}(t) = \frac{L}{G}\dot{\tau}(t) + M(\beta\Psi + \beta^2\tau), \quad (6)$$

where $M$ is the GB mobility, $\tau$ is the shear stress, and $G$ is the shear modulus.

For the special case where the disconnections are perfect steps ($\mathbf{b}_i = 0$), such that $\beta = 0$, then $\dot{H} = M\Psi$ and $\dot{D} = (L/G)\dot{\tau}$. GB migration is then decoupled from $\tau$ and $D$. In the remainder of the discussion, we implicitly assume that $\beta \neq 0$ (although this presents no problem). We now consider two cases: stress-controlled migration and displacement-controlled migration.

Fixed Stress, $\tau = \tau^0$: First, we consider a constant stress or traction applied at the ends of the sample. From Eqs. (5) and (6), $\dot{D} = M(\beta\Psi + \beta^2\tau^0)$ and $\dot{H} = M(\Psi + \beta\tau^0)$. The GB migrates to the top of the cell and the top of the cell displaces both at constant rates. For the unconstrained (free surface) boundary condition

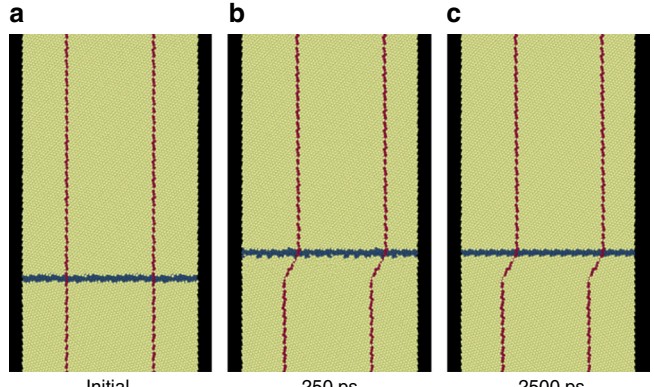

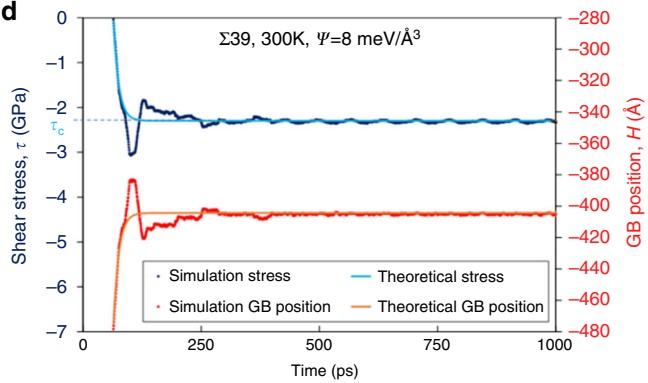

**Fig. 6** Shear-coupling in a symmetric tilt GB with fixed ends. **a–c** Time sequence of a bicrystal simulation identical to that in Fig. 5 except that the top and bottom surface of the simulation cell are fixed (do not translate). **d** Shear stress $\tau$ and GB position H from the same simulation, compared with Eq. (7) (theoretical stress) and (8) (theoretical GB position) from the single-mode migration model, using measured values of mobility, $\beta$, and $\Psi$. The overshoot of the simulation data vs. the theoretical prediction is associated with the elastic response of the large simulation cell

($\tau^0 = 0$, see Fig. 5), $\dot{H} = M\Psi$ and $\dot{D} = \beta M\Psi$. This is the commonly used synthetic driving force simulation approach[31].

Fixed displacement rate, $\dot{D} = \dot{D}^0$: Many studies of shear-coupled GB migration incorporate a fixed displacement rate $\dot{D}$[21]. To model this, we rewrite Eq. (6) as $\dot{\tau} = (G/L)[\dot{D}^0 - M\beta(\Psi + \beta\tau)]$, and integrate:

$$\tau(t) = \frac{\dot{D}^0}{M\beta^2} - \frac{\Psi}{\beta} + \left(\tau^0 + \frac{\Psi}{\beta} - \frac{\dot{D}^0}{M\beta^2}\right) e^{-\frac{GM\beta^2}{L}t}. \quad (7)$$

Substituting Eq. (7) into Eq. (5) and integrating with respect to time yields (for $H_0 = 0$)

$$H(t) = \frac{\dot{D}^0}{\beta} t - \frac{L}{G\beta^2}\left(\tau^0\beta + \psi - \frac{\dot{D}^0}{M\beta}\right)\left(e^{-\frac{GM\beta^2}{L}t} - 1\right) \quad (8)$$

In the constrained simulations (Figs. 5 and 6), $\dot{D} = 0$ and $\tau^0 = 0$. In steady state ($t \rightarrow \infty$), this approaches

$$\tau^\infty = \frac{\dot{D}^0}{M\beta^2} - \frac{\Psi}{\beta} = -\frac{\Psi}{\beta} \quad (9)$$

$$H^\infty(t) = \frac{\dot{D}^0}{\beta} t + \frac{L}{G\beta^2}\left(\Psi - \frac{\dot{D}^0}{M\beta}\right) = \frac{L}{G\beta^2}\Psi. \quad (10)$$

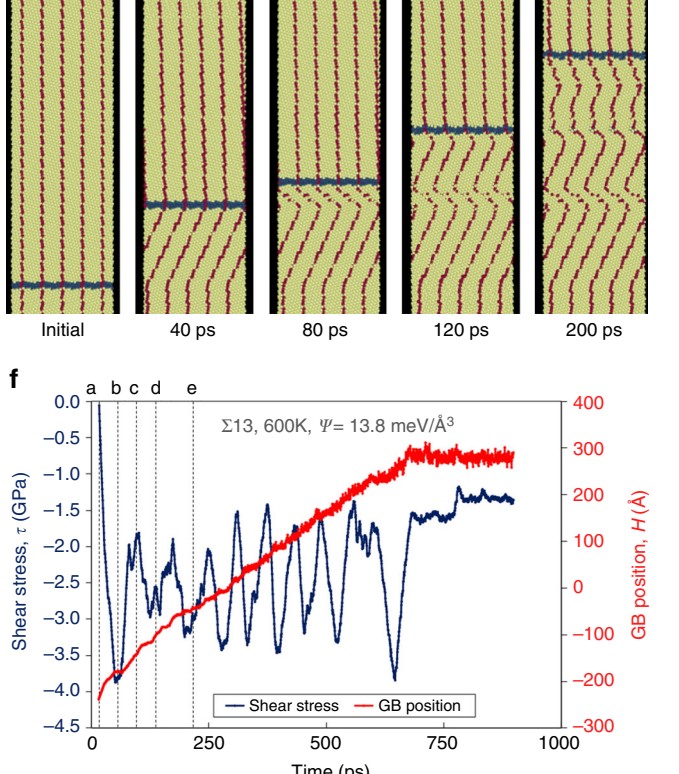

**Fig. 7** Mode-switching in a symmetric tilt GB with fixed ends. **a–e** Time sequence of a $\Sigma13$ [111] $(3\bar{4}1)$ symmetric tilt GB under the same constraint conditions as Fig. 6. The temperature was 600K and the driving force was 13.8 meV Å$^{-3}$. Unlike Fig. 6, the GB does not stagnate, but instead continues migrating. The zig-zag pattern in the fiducial mark indicates repeated changes in shear-coupling mode. **f** GB position and shear stress $\tau$ vs. time for this simulation, indicating continuous GB migration and oscillating stress. Alphabetical labeling in **f** indicates the corresponding time in **a–e**

The GB travels a finite distance before stopping with a steady-state stress, consistent with the observations in Fig. 6. The time evolution of $\tau$ and $H$ agrees with simulation results shown in Fig. 6d (solid, colored lines). Here, we have used independently measured values of $\beta$, $G$, $M$, and $L$. However, while this approach is consistent with the constrained $\Sigma39$ simulation results in Fig. 6, it fails to describe the zig-zag motion in Fig. 7, which indicates disconnection mode-switching. To model this behavior, we must consider multiple coupling modes.

**Multiple disconnection modes**. To understand the zig-zag motion of the $\Sigma13$ GB (Fig. 7), we consider the thermal nucleation of disconnections along an initially flat GB within a periodic simulation cell. We further assume that nucleation is slow compared with the migration and annihilation of disconnections such that the GB effectively remains flat. These are reasonable for the relatively narrow, periodic bicrystal simulations here.

For each GB, there is an infinite set of possible disconnections $(\mathbf{b}_i, h_i)$. The barrier to forming a disconnection pair of type $i$ depends on the energy required to form the disconnection pair itself, the interactions between disconnections (together we label these as $E_i$), and the driving force to separate the two disconnections $f_i$[55]. The nucleation barrier is $E_i - f_i$, where

$$E_i = 2\gamma_S|h_i| - \frac{G}{2\pi}\frac{1}{1-\nu}b_i^2 \ln\left[\sin\left(\frac{\pi\delta_0}{w}\right)\right]. \quad (11)$$

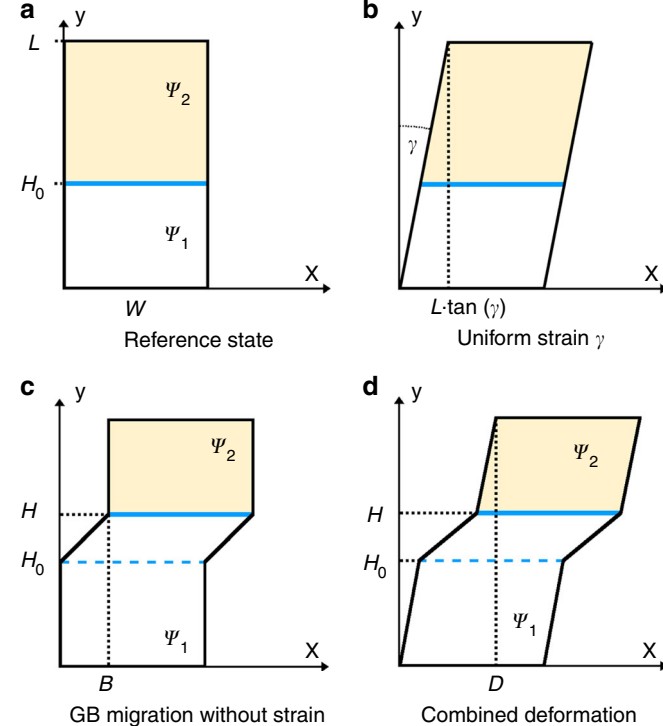

**Fig. 8** Schematic represenation of the elastic model with reference configuration **a**. The total deformation can be separated into the elastic component **b** and the plastic component **c**. The total deformation is depicted in **d**.

The first term is the excess energy associated with the GB step and the second accounts for the dislocation core energy and energy required to separate the disconnections. $\nu$ is the Poisson ratio of the material, $w$ is the length of the GB (periodic unit cell), and $\delta_0$ is the dislocation core radius. The contribution to the nucleation barrier due to the driving force on the GB is $f_i = w(h_i\Psi + b_i\tau)/2$. $E_i$ and $f_i$ are normalized by the thickness of the bicrystal.

The nucleation rate of a disconnection pair of type $i$ is proportional to $e^{-(E_i-f_i)/kT}$. However, disconnection pairs come in equal, opposite sets $\pm(\mathbf{b}_i, h_i)$. We can write $\dot{B}$ and $\dot{H}$ in terms of the nucleation rates of all disconnection pair types as follows:

$$\dot{B} = \omega\sum_i b_i\left(e^{-\frac{E_i-f_i}{kT}} - e^{-\frac{E_i+f_i}{kT}}\right) \quad (12)$$

$$\dot{H} = \omega\sum_i h_i\left(e^{-\frac{E_i-f_i}{kT}} - e^{-\frac{E_i+f_i}{kT}}\right), \quad (13)$$

where $\omega$ is an attempt frequency and the macroscopic shear-coupling parameter is $\beta = \dot{B}/\dot{H}$. If one disconnection mode ($i = 0$) dominates ($E_0 \ll E_i$ for $i \neq 0$ or $T \to 0$), then $\beta \to \beta_0 = b_0/h_0$. This is single-mode coupling.

For $f_i \ll kT$, we can expand the exponentials in Eqs. (12) and (13) and substitute $f_i = w(h_i\Psi + b_i\tau)/2$:

$$\dot{B} = \frac{\omega w}{kT}\left(\tau\sum_i b_i^2 e^{-\frac{E_i}{kT}} + \Psi\sum_i h_i b_i e^{-\frac{E_i}{kT}}\right) = K_{11}\tau + K_{12}\psi \quad (14)$$

$$\dot{H} = \frac{\omega w}{kT}\left(\tau\sum_i h_i b_i e^{-\frac{E_i}{kT}} + \Psi\sum_i h_i^2 e^{-\frac{E_i}{kT}}\right) = K_{21}\tau + K_{22}\Psi, \quad (15)$$

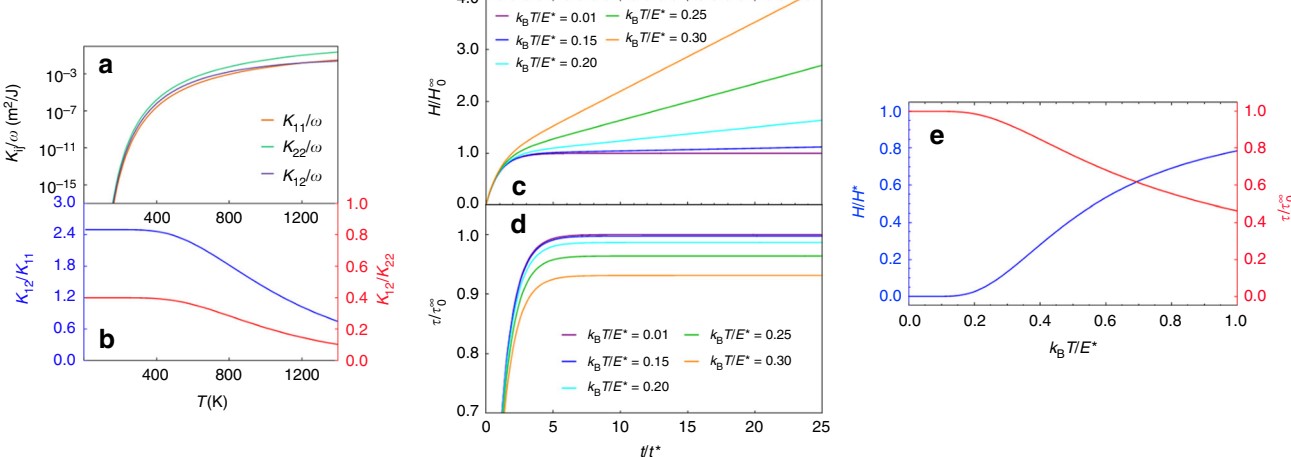

**Fig. 9** Multi-mode Model. **a** Temperature dependence of the Onsager Coefficients $K_{ij}$ for the $\Sigma13$ [001] (510) symmetric tilt GB in Ni, **b** the ratios $K_{12}/K_{11}$ and $K_{12}/K_{22}$ using Eq. (11). **c** $\tau(t)/\tau_0^\infty$ and **d** $H(t)/H_0^\infty$ as a function of time for various temperatures in the two-mode model. **e** $\lim_{t\to\infty} \tau(t)/\tau_0^\infty$ and $\lim_{t\to\infty} \dot{H}(t)/\dot{H}^*$ for the two-mode model as a function of temperature. $t^*$ is the time constant introduced in Eq. (17)

where $K_{ij}$ may be viewed as Onsager coefficients and $K_{12} = K_{21}$ (we confirmed that $\dot{B}$ and $\dot{H}$ are near linear functions of $\Psi$ via independent simulations[31]. These results, in principle, include the effects of all possible disconnections and describe the full temperature-dependent behavior of any GB.

While the summands in $K_{11}$ and $K_{22}$ are positive-definite, those in $K_{12}$ are not. We therefore, expect the diagonal terms to dominate at high temperature. In this limit, for stress-driven GB migration ($\Psi = 0$, $\tau \neq 0$) $\beta = \dot{B}/\dot{H} \to \infty$, corresponding to perfect sliding. For migration driven by an energy density difference between two grains ($\tau = 0$, $\Psi \neq 0$), $\beta \to 0$, corresponding to GB migration with zero net shear deformation. The explicit temperature-dependence of $K_{11}$, $K_{22}$, and $K_{12}$ using known values of $(b_i, h_i)$ for a $\Sigma13$[001](510) symmetric tilt GB and material properties for the Ni potential used above[57] is shown in Fig. 9a, b. These trends may be considered generic for all GBs.

We now consider GB migration under stress-controlled and displacement-controlled conditions for the multi-mode case. Referring to Fig. 8d, we note that

$$\dot{B} = \dot{D} - (L/G)\dot{\tau}, \qquad (16)$$

where $\dot{B}$ depends on temperature, stress, $\Psi$, simulation dimensions, and includes all possible disconnections.

Fixed Stress, $\tau = \tau^0$: For fixed stress, Eqs. (15) and (16) imply $\dot{D} = K_{11}\tau^0 + K_{12}\Psi$ and $\dot{H} = K_{12}\tau^0 + K_{22}\Psi$. This resembles the single-mode case; the GB migrates and the top of the cell translates, both at constant velocity. However, if the off-diagonal term $K_{12}$ vanishes at high temperature, either perfect sliding (for $\psi = 0$, $\tau \neq 0$) or GB migration without shear-coupling (for $\tau = 0$, $\psi \neq 0$) occurs. This multi-mode analysis explains why $\beta$ is a function of temperature and driving force; this is in contrast to conventional (single mode) shear-coupling for which $\beta$ is constant.

Fixed displacement rate, $\dot{D} = \dot{D}^0$: This final case corresponds to Fig. 7. Here, the distinction between single-mode and multi-mode migration becomes even more important; as the stress evolves, so does the relationship between $\dot{D}$ and $\dot{H}$. Combining Eqs. (15) and (16) and integrating with respect to time yields

$$\tau = \tau^0 e^{-t/t^*} + \frac{\dot{D}^0 - K_{12}\Psi}{K_{11}}\left(1 - e^{-t/t^*}\right), \qquad (17)$$

where $t^* = L/(GK_{11})$. If $\dot{D}^0 = 0$ and $\tau^0 = 0$ (Fig. 7),

$$H(t) = \Psi\left[\left(K_{22} - \frac{K_{12}^2}{K_{11}}\right)t + \frac{L}{G}\left(\frac{K_{12}}{K_{11}}\right)^2\left(1 - e^{-t/t^*}\right)\right].$$

As $t \to \infty$,

$$\tau^\infty = -\frac{K_{12}}{K_{11}}\Psi, \qquad \dot{H}^\infty = \left(K_{22} - \frac{K_{12}^2}{K_{11}}\right)\Psi$$

$$H(t) = \left[\left(K_{22} - \frac{K_{12}^2}{K_{11}}\right)t + \frac{L}{G}\left(\frac{K_{12}}{K_{11}}\right)^2\right]\Psi.$$

Rather than stagnating, the GB will migrate at a constant rate at late times.

At high temperature, the terms containing $K_{12}$ vanish and the boundary migrates at a constant velocity $\dot{H} = K_{22}\Psi$ with no stress accumulation. Here, $K_{22}$ describes the conventional mobility of the GB. At low-temperatures, where a single mode ($\mathbf{b}_0$, $h_0$) dominates, we recover Eqs. (9) and (10):

$$\tau_0^\infty = -\frac{K_{12}}{K_{11}}\Psi = -\frac{\Psi}{\beta_0}$$

$$H_0^\infty = \frac{L}{G}\left(\frac{K_{12}}{K_{11}}\right)^2 = \frac{L}{G\beta_0^2}\Psi.$$

Even when the GB appears to stagnate at low (finite) temperature, there will be a small, constant velocity. However, Fig. 9 suggests that this velocity will be extremely small. We associate this velocity with the rare nucleation of a disconnection pair with a high barrier.

The general case is difficult to address analytically, but we can examine a case where migration is controlled by two types of disconnections ($\mathbf{b}_0$, $h_0$) and ($\mathbf{b}_1$, $h_1$) (i.e., an intermediate temperature). For example, consider a GB for which $b_0 = b$, $h_0 = b$, $E_0 = E$, $b_1 = -b$, $h_1 = b$, and $E_1 = 2E$. The dimensionless quantities $\tau/\tau_0^\infty$, $H/H_0^\infty$, and $\dot{H}/\dot{H}^*$ (where $\dot{H}^* = K_{22}\Psi$, the velocity when $\tau = 0$) are independent of $\omega$, $\Psi$, the system dimensions, and the specific choice of $b$, depending only on the relative values of $b_i$, $h_i$, $E_i$, and $T$. The time evolution of $\tau/\tau_0^\infty$ and $H/H_0^\infty$ for various temperatures is given in Fig. 9c, d and the steady-state values $\tau^\infty/\tau_0^\infty$ and $\dot{H}^\infty/\dot{H}^*$ as a function of temperature are given in Fig. 9e. There is a range of low temperatures for which the GB very-nearly stagnates with a stress of $\tau^\infty = \tau_0^\infty$. As temperature and (by extension) the ($\mathbf{b}_1$, $h_1$) nucleation rate increases, $\tau^\infty$ decreases and $\dot{H}$ increases.

We now apply the disconnection model directly to the GBs simulated in Figs. 6 and 7. The barriers (Eq. (11)) depend on the spacing between nuclei ($w$ in the limit of a narrow, periodic simulation) and material properties (see Supplementary Note 3). For the simulations in Figs. 6 and 7, we can infer the dominant coupling modes for each GB based on the analyses of[55] and[39]. For the $\Sigma 39$ GB in the arrested case (Fig. 6), the disconnection modes correspond to $b = 2\sqrt{3}na_{dsc}$ and $h = (6n + 39j)a_{dsc}$, where $n$ and $j$ are integers and $a_{dsc} \equiv a_0/(2\sqrt{78})$, where $a_0$ is the lattice constant. The two disconnections ($\mathbf{b}_i$, $h_i$) with the smallest $E_i$ corresponding to $(n, j) = (1, 1)$ and $(2, 2)$. Both modes correspond to the same $\beta = 1/\sqrt{3} \approx +0.58$, even though they correspond to different $E_i$. Since both modes have the same sign of $\beta_i$, activation of both would not relax the stress accumulation and even higher $E_i$ modes would be necessary to facilitate further migration. On the other hand, for the $\Sigma 13$ GB (which exhibits switch-back behavior), the allowed disconnections modes correspond to $b = 6na_{dsc}$ and $h = (6\sqrt{3}n + 13\sqrt{3}j)a_{dsc}$. Those with the smallest $E_i$s correspond to $(n, j) = (1, 1)$ and $(1, -1)$. In this case, the $E_i$ gap is much smaller than in the $\Sigma 39$ GB case and the corresponding modes have $\beta_i$ values of opposite sign; i.e., $\beta = 1/\sqrt{3} \approx 0.58$ and $\beta = -6/(7\sqrt{3}) \approx 0.50$, respectively. This explains why the $\Sigma 13$ GB readily migrates by alternating between two disconnection modes, while the $\Sigma 39$ GB stagnates.

Neither curvature flow nor ideal shear-coupling completely describe the general nature of GB migration. However, the disconnection model of GB migration is able to explain both GB stagnation and the observed switch-back behavior (as well as everything in between). The main difference between the $\Sigma 13$ and $\Sigma 39$ GBs in Figs. 6 and 7 is the availability of disconnection modes with relatively low $E_i$ and $\beta_i$ values of opposite sign to the lowest-$E_i$ mode. This enables the $\Sigma 13$ to access a coupling mode that relaxes stresses generated by the first coupling mode and facilitates long-distance GB migration. Figure 9c–e show that while some mode switching may occur at any temperature, the degree to which mode switching for each particular GB is important depends on temperature. These results clearly demonstrate that even a very-simple two-mode model is capable of describing this rich behavior.

**Idealized polycrystalline microstructure revisited**. We can apply our conclusions thus far to the idealized microstructure simulation in Fig. 3. Unlike the GBs in Figs. 6 and 7, the relevant GBs in the idealized microstructure simulation are asymmetric-tilt boundaries. The migration mechanisms of asymmetric tilt GBs are more complicated than those of symmetric tilt GBs, and the details of how they will behave under general conditions is still an active subject of study[58–60]. However, we can still apply the same types of bicrystal simulations (free and/or constrained) to qualitatively infer whether their behavior under constraint is consistent with our observations in Fig. 3.

We characterize each of the GBs in the idealized microstructure (Fig. 3a) using bicrystal simulations under both free and fixed-end conditions. The results are given in Supplementary Note 2. As Supplementary Figs. 1 and 2 show, all four GBs in the free-end simulations exhibit shear-coupling even at the simulation temperature of $0.85T_m$ and all four GBs show similar velocities (and do not exhibit stick/slip behavior)[31].

Figure 10a shows the direction of GB migration (assuming Grains A and B shrink due to capillarity) and sense of $\beta$, as well as the expected rotation/stress-generation of each grain in the ideal microstructure, based on Supplementary Fig. 2. Figure 10b shows the same results based on the actual simulation observations from Fig. 4. The sense of the shear stress that develops in Grain A (Fig. 4a) is consistent with the signs of $\beta$ as measured from

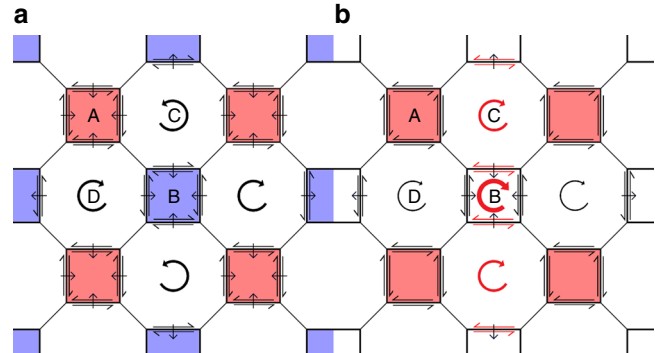

**Fig. 10** Idealized Microstructure Predicted and Observed Behavior. **a** Predicted shear-coupling sense (arrows parallel to each GB), stress (grain shading), and rotation (circular arrows) according to bicrystal simulations for the idealized microstructure. **b** Same as in **a** but based on the actual microstructure evolution simulations (Fig. 4). Red arrows denote behavior in **b** that contradicts the predictions in **a**. Red and blue shading denote shear stress of opposite sign

Supplementary Fig. 2 (*cf.* Fig. 10a, b). The stress accumulation (Fig. 4a) and GB stagnation (Fig. 3) can be considered by the analogy with the arrested GB migration in Fig. 6. However, these same measurements suggest (Fig. 10a) that Grain B should also become stressed. The observed rotations in Grains B and C imply that the GB that separates them (GB 4) is migrating with a coupling mode of sign opposite to that implied by Supplementary Fig. 2. Therefore, ideal shear-coupling cannot describe how this grain behaves, just as it was insufficient to describe the mode-switching behavior in Fig. 7.

Supplementary Fig. 3 shows the results of simulations of each of the four GBs in the idealized microstructure, performed under constraints (fixed-end) as in Figs. 6 and 7. Supplementary Fig. 3 shows that GBs 1–3 exhibit stress accumulation and arrested migration behavior, similar to the simulation in Fig. 6. Critically, GB 4 (Supplementary Fig. 3d) is the only GB that migrates a long distance in the constrained simulations; this clearly indicates mode-switching (with the concomitant stress oscillation). Grain B shrinks because GB 4 exhibits mode-switching, while the other GBs do not. Effectively, GB 4 migrates and slides (i.e., switches between modes with opposite-sign $\beta_i$), facilitating grain rotation without stress accumulation (relaxing the stress associated with the migration of GB 3).

The disconnection model of GB migration, incorporating the effects of constraints endemic to all microstructures, allows us to understand the microstructure evolution in Fig. 3. This is significant because initial analysis suggested a wide range of previously puzzling events (GB migration stagnation, grain rotation, stress accumulation, and the different behaviors of two grains with identical grain shape). Analysis of the migration of an GB in a real microstructure is possible, but the complexity of general GBs and microstructures makes this formidable for an entire microstructure.

## Discussion

The results presented above suggest that shear-coupling is an intrinsic behavior of grain boundary migration. The present results are a strong support for the speculation that this statement is true for all grain boundaries under all conditions and it is simply a consequence of the underlying mechanism by which GBs migrate. This should apply at low temperatures, where shear-coupling is obvious, and at high temperatures, even when GBs appear to slide. It is important when GB migration leads to stress generation and when it does not. It is not surprising, therefore, that shear-coupling

can lead to complex effects during grain growth in a polycrystal, and that grain rotation, stress generation (and potentially defect generation), and grain growth stagnation are all inter-related and dependent on shear-coupling and disconnection mode selection. This suggests a much richer grain boundary dynamics/kinetics than is commonly assumed in conventional curvature-driven GB migration. This richness is in addition to that arising via non-disconnection related phenomena, such as grain boundary torque, crystal plasticity (and its interaction with GBs), and grain boundary diffusional accommodation, each of which may occur on separate time and length scales.

Real microstructures necessarily imply (at least) two types of constraints on GB migration and shear-coupling. First, two grains cannot simply shear relative to one another within a micro-structure without severe mechanical consequences (stress generation). Second, since grain boundaries are necessarily delimited by triple (and higher order) junctions, shear-coupling must generate stress at these junctions. Hence, constraint plays a major role in shear-coupled GB migration and, in turn, grain growth.

While shear-coupling necessarily implies significant stress generation, nature endows microstructures with a myriad of possibilities for relieving the stress associated with GB migration. These include extrinsic mechanisms such as twinning and dislocation generation (and more generally, plasticity within the grains)—as observed in our simulations (Fig. 1b). It also includes intrinsic mechanisms, largely associated with the availability of multiple disconnection modes—nature's choice of which depends on the stresses generated and temperature. If the stresses get too large and no relaxation mechanism is available, such coupling and constraints can lead to the cessation of GB migration.

These observations also motivate the need to reconsider the notion of grain boundary mobility. GB migration is not a monolithic thermally-activated process, but rather an aggregate of thermally-activated disconnection nucleation and migration events. As disconnection selection and nucleation are sensitive to the environment (e.g., stress and other driving forces) in which a GB migrates, so too the GB mobility varies not only between GBs, but for a given GB as it migrates within the material.

While the constraint associated with neighboring grains may be less pronounced in microstructures with large average grain sizes, grain boundaries must also migrate further to achieve the same fractional increase in grain size. The development of stresses associated with shear-coupling will, therefore, be essentially grain-size independent. This is not true of stress-relaxation mechanisms. It is our expectation that the macroscale effects of stress-coupling will be more important with decreasing grain size—where operative relaxation mechanisms are more limited. We also note that even in a microstructure with a large mean grain size, GBs bounding smaller grains will behave differently from those bounding larger ones.

We note that grain boundaries themselves have degrees of freedom that are not associated with disconnections at all. These include the large range of metastable structures associated with grain boundaries[61], as well as compositional degrees of freedom, which may modify disconnection dynamics.

Ultimately, the conventional theory of grain growth is expected to still be functional on the macroscopic scale-largely because of stress relaxation effects (e.g., at high temperature, GBs may effectively slide with relatively little shear-coupling). On the other hand, the conventional assumption that GB mobility is only a function of macroscopic GB bicrystallography is particularly simplistic. Conventional grain growth theory must be viewed as a simplistic model that does not account for the fact that real materials are crystalline and crystallinity imposes constraints on how GBs move. Simply put, polycrystals are not soap froths, even

though that analogy has served us in good standing for over a half century[62].

The present work examines some of the implications of crystal structure on the evolution of polycrystalline microstructures. Of course, this falls within the framework of a wide-range of work on GB migration and shear-coupling in recent years. However, it should be viewed as an initial discussion with many implicit assumptions and approximations. The present work has also shown the considerable complexity in trying to apply our present understanding of grain boundary migration to polycrystalline systems. This complexity is enormous (given the 5 macroscopic degrees of freedom of GB macro-crystallography, the inter-connectedness of GB networks, the presence of triple and higher order junctions, GB metastability, etc). The challenge is to take this type of microscopic mechanistic theory and deduce an effective equation of motion that can be applied to predict overall GB migration while retaining only the essential ingredients (approximating those that have little macroscopic effect). Borrowing (liberally) from Einstein: "grain boundary dynamics should be made as simple as possible, but no simpler" (e.g., see ref. [63]).

## Methods

**Polycrystal grain growth simulations**. In polycrystal grain growth simulations, initial microstructure can have a profound effect on subsequent evolution. In the polycrystal simulation presented here, the initial structure was created by generating a steady-state, curvature flow, polycrystal microstructure[64], assigning the orientation of each grain at random, and generating the associated face centered cubic (FCC) atomic lattice within the structure. This was done instead of the more common method of generating a voronoi tesselation from a poisson distribution of points, which produces flat GBs, unrealistic triple junction angles, and grain size distributions inconsistent with grain growth microstructures[65].

After the initial configuration was generated, atoms were removed which were closer than 60% the equilibrium 0 K nearest neighbor distance and the atomic configuration was relaxed at $T = 0$ K. Finally, the system was annealed for 100 ps at 300 K and then relaxed again. This approach was used to remove any artifacts of the process that generated the initial configuration and separate phenomena associated with grain growth from those related to relaxing the as-constructed, high-energy GB structures.

The polycrystal simulation cell had edge length $W \approx 400$ Å and the simulation consisted of approximately 5,000,000 atoms. The polycrystal was annealed in an $NPT$ ensemble (Nose′–Hoover thermostat) at $0.85T_m$ and zero external pressure. The temperature was chosen high enough to facilitate significant grain growth in a reasonable computation time, but low enough to prevent GB pre-melting. The simulation time was limited to 2.5 ns to prevent individual grains from spanning the simulation cell. Mean grain size was calculated as $\ell = W/N^{1/3}$, where $N$ is the number of grains and $W$ is the simulation cell width.

**Bicrystal simulations**. In both the free-end (Fig. 5) and fixed-end (Fig. 6) simulations of the $\Sigma 39$ symmetric-tilt bicrystal, the simulation cell had dimensions $1,914$ Å × $72$ Å × $18$ Å in the directions perpendicular to the GB ($L$), parallel to the GB ($w$), and perpendicular to the plane of Fig. 5. (The GB position $H$ in these figures is referenced to the center of the simulation cell.) The driving force in these simulations is an energy density difference between two grains of different orientation[54] of magnitude 8.0 meV Å$^{-3}$ (energy density difference between the two grains). The simulations were performed with the same EAM nickel potential as in the previous simulations[57], under an $NPT$ ensemble with zero external stress, at 300 K. In the constrained simulation of the $\Sigma 13$ symmetric tilt bicrystal (Fig. 7), the simulation cell dimensions were 956 Å × 16 Å × 18 Å.

The bicrystal simulations of the GBs in the idealized microstructure (Fig. 3 were performed at $0.85T_m$ (the same temperature as the simulation in Fig. 3, and with $\Psi = 4.4$ meV Å$^{-3}$. GB energies were computed by relaxing the atomistic configuration at 0 K. GB mobilities ($M$) and shear-coupling factors $\beta$ for each of the bicrystals were measured using a synthetic driving force potential method[54] with an energy difference between the two grains of 4.4 meV Å$^{-3}$ at the same temperature as the "simple" microstructure simulations. The results are given in Supplementary Table 2. All bicrystal simulations are periodic in the horizontal ($\hat{x}$) direction as well as the direction normal to the viewing plane ($\hat{z}$).

**Data availability**. The data that support the findings of this study are available from the authors on reasonable request, see author contributions for specific data sets.

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

## Acknowledgements

S.L.T. acknowledges support from the Department of Education GAANN program, grant number P200A160282. D.J.S. and J.H. gratefully acknowledge partial support of the NSF Division of Materials Research through Award 1507013. P.K.P. and D.J.S. are grateful for partial support through Penn's NSF MRSEC under Award No. DMR-112090.

## Author contributions

S.L.T. and J.H.: Performed and analyzed the computer simulations. S.L.T., K.C., J.H., D.J.S. and P.K.P.: Constructed the theoretical model. D.J.S.: Provided the overall design of this research project.

## Additional information

**Competing interests:** The authors declare no competing financial interests.

