## [Peer Review File · Nature Communications]

Reviewers' Comments:

Reviewer #1:

Remarks to the Author:

The authors present a particularly well written study that for the first time, reconcile discrepancies between commonly accepted views on grain growth and the process of shear coupled boundary migration. Further, the authors elegantly use simple transition state theory argument, with a theory of defects resolved at the interface level to rationalize temperature driven transitions in the dominant process leading to interface motion. Further, the atomistic work which motivates the theoretical development are described with great precision.

Overall, the reviewer strongly feels that the study proposed will be very influential in the materials science community as prior to this work there was no means to reconcile interfacial defect theory - as a static approach- and atomistic simulations of grain growth.

Minor comment: PSI should be defined in the body of the paper and not just in the supplementary material.

Reviewer #2:

Remarks to the Author:

The authors have done an impressive job in demonstrating the important role of shear-coupled grain boundary migration in grain growth using large-scale polycrystal and bicrystal atomistic simulations. Though there are previous literature showing shear-coupling in grain growth experiments, not to the quantitative, atomistic and theoretical extent shown by the present work, which significantly advances our understanding of grain growth.

The manuscript is well-written, with solid methodology, in-depth discussion, appropriate references and clear figures. The proposed theoretical model is new and well-corroborated by the atomistic simulations. The results provide reasonable and interesting explanations to phenomena such as grain boundary migration stagnation, grain rotation and stress accumulation in grain growth. I recommend the publication of this work in Nature Communications once the concerns listed below are addressed.

The authors showed that the disconnection nucleation barrier gap between the lowest- E_{i_i} mode and the lowest- E_{i_i} mode of opposite sign determines coupling behaviors of $\Sigma 13$ and $\Sigma 39$ GBs under constraint, which indicates the importance of GB character. Since GBs with a small barrier gap are likely to switch mode and facilitates grain growth. I am curious to know the percentage of the mode-switching GBs in polycrystal grain growth simulations. Could the authors give the readers an idea the probability of encountering a small-barrier gap GB within the five macroscopic degrees of freedom in the GB configurational space and how to find one? If this is too general, the authors can limit their discussion to $[100]$, $[110]$ or $[111]$ symmetric tilt GBs.

Line 255: Should the word "construction" be replaced by "constriction"?

Line 370: Citations should be provided for the known values of (b_i, h_i) for the $\Sigma 13[001](510)$ GB.

Line 526 "Discussion": The sentence "The results presented above suggest that shear-coupling is an intrinsic behavior of all grain boundaries under all conditions" is a big assertion. Though it is likely to be true considering the dislocation nature of grain boundaries, I don't think such claim is appropriate before testing it on all types of grain boundary taking into account all the macroscopic and microscopic degrees of freedom. A slight change of tone would be good, like ".....shear-coupling might be an intrinsic.....".

Line 657 "Methods": Does the sentence "...individual grains from spanning the simulation cell...." mean "facilitate individual grains from..." or "prevent individual grains from..."?

Line 664 "Methods": $1914 \times 72 \times 18 \text{ \AA}$ should be $1914 \times 72 \times 18 \text{ \AA}^3$. Similar places in the main text and supplementary information should also be changed.

Moreover, I noticed that there are four videos, which have neither been referred in the main text nor been explained in the supplementary information.

Reviewer #3:

Remarks to the Author:

This paper marks a significant milestone in understanding the behavior of grain boundaries in metals - though it is certainly not the end of the journey.

For several decades, research on grain boundaries has proceeded along largely parallel tracks. Grain boundaries in polycrystals are assumed to behave according to simple rules, mostly based on capillarity considerations. Individual grain boundaries, however, are treated in the context of a hierarchy of crystal lattice defects, in which they have individual properties and exhibit behaviors such as linked migration and sliding, which is the focus of much of the present paper.

Until now, the complexity of the linkages between interfaces in a polycrystal has defied attempts to reconcile the two approaches, although various ideas for overcoming some of the challenges have been discussed - mostly over a lot of beer.

This paper makes explicit observations of the effects of individual grain boundary structures in the context of a polycrystal, during grain growth. It goes further, by comparing the behavior of some individual boundaries in cases where they are unconstrained by polycrystalline connectivity with cases where the constraints are crudely modeled. The induction of multiple modes of disconnection-mediated grain boundary motion in constrained cases confirms what had often been postulated zymurgically, but never analyzed in detail up to now. This is a very nice use of computer modeling to find a result that could never be accessed experimentally.

The paper suggests a rationalization for the formation of annealing twins, including some of the observed peculiarities of that phenomenon. This is a long-standing puzzle that has only had very unsatisfactory explanations up to now, more-or-less on the order of "nature abhors an untwined crystal."

Obviously I think this paper is a significant step forward in understanding polycrystalline behavior. There are, however, a few very minor concerns:

First, in terms of nomenclature, I think that in the context of this paper, some fine distinctions are helpful, even if they are not commonly made elsewhere. In particular, the term "anisotropy" (of grain boundary properties) is applied a little loosely. Anisotropy refers to the change in properties of a grain boundary as its plane is rotated (at fixed misorientation.) The variation of energy with misorientation is really NOT anisotropy. In the wonderland of theory, a boundary can have properties that are isotropic AND misorientation-dependent, and I think that the distinction is worth maintaining.

In the case of grain rotation, for example, the two explanations are migration-induced shear (or vice versa) and misorientation-dependent energy, but NOT anisotropy.

On the topic of grain rotation, it is clear, I believe, that all of the observed cases presented in this

paper are migration-induced shear. Among the tell-tale signs, the rotation is unidirectional, while it would be expected to be more like a rotational form of Brownian motion for the case of misorientation-dependent energy.

The authors do not imply that grain rotation cannot be driven by misorientation-dependent energy, but perhaps they might wish to make that clear, along with the fact that it is unlikely to be observed over the time and length scales studied in the current work. Indeed, this is the only major challenge with interpreting the results presented here: they relate to competing processes that must certainly have differing temperature, time and length scale dependences, at the individual boundary level as well as at the level of the polycrystalline assembly. The specific results, must therefore be considered with some caution as to their application at different length scales, timescales and temperatures.

That said, the findings of this work are exciting, and the phenomena that have been discovered are likely to be important aspects of understanding polycrystalline behavior going forward. It is, to the best of my knowledge, the first time that grain boundary structure considerations have been applied to polycrystalline behavior in a systematic and logical way.

Response to Reviewers

Reviewer #1 (Remarks to the Author):

The authors present a particularly well written study that for the first time, reconcile discrepancies between commonly accepted views on grain growth and the process of shear coupled boundary migration. Further, the authors elegantly use simple transition state theory argument, with a theory of defects resolved at the interface level to rationalize temperature driven transitions in the dominant process leading to interface motion. Further, the atomistic work which motivates the theoretical development are described with great precision.

Overall, the reviewer strongly feels that the study proposed will be very influential in the materials science community as prior to this work there was no means to reconcile interfacial defect theory -as a static approach- and atomistic simulations of grain growth.

Minor comment: PSI should be defined in the body of the paper and not just in the supplementary material.

We appreciate the accolades for our manuscript from Reviewer #1. Following his/her request, we now put the conceptual definition of Ψ on lines 217 and 251 in the main text. We also refer to the reference from which this comes (and a detailed description) at this point.

Reviewer #2 (Remarks to the Author):

The authors have done an impressive job in demonstrating the important role of shear-coupled grain boundary migration in grain growth using large-scale polycrystal and bicrystal atomistic simulations. Though there are previous literature showing shear-coupling in grain growth experiments, not to the quantitative, atomistic and theoretical extent shown by the present work, which significantly advances our understanding of grain growth.

The manuscript is well-written, with solid methodology, in-depth discussion, appropriate references and clear figures. The proposed theoretical model is new and well-corroborated by the atomistic simulations. The results provide reasonable and interesting explanations to phenomena such as grain boundary migration stagnation, grain rotation and stress accumulation in grain growth. I recommend the publication of this work in Nature Communications once the concerns listed below are addressed.

We appreciate the kudos from Reviewer #2.

The authors showed that the disconnection nucleation barrier gap between the lowest- E_i mode and the lowest- E_i mode of opposite sign determines coupling behaviors of $\Sigma 13$ and $\Sigma 39$ GBs under constraint, which indicates the importance of GB character. Since GBs with a small barrier gap are likely to switch mode and facilitates grain growth. I am curious to know the percentage of the mode-switching GBs in polycrystal grain growth simulations. Could the authors give the readers an idea the probability of encountering a small-barrier gap GB within the five macroscopic degrees of freedom in the GB configurational space and how to find one? If this is too general, the authors can limit their discussion to $[100]$, $[110]$ or $[111]$ symmetric tilt GBs.

The Reviewer asks an interesting question here, “how common is disconnection mode switching during polycrystalline grain growth?” Unfortunately, this question is difficult to address, largely because the answer is simply “it depends.” To be a bit less glib, whether a GB mode switches or not is determined not just by the type of GB (5 macroscopic degrees of freedom) but also by the environment. That is, it depends on the temperature and what is going on in the surrounding GBs and the triple junctions. So, instead of answering this specific question in the text, we address it in the final paragraph of the “Shear-Coupled Bicrystal Simulations” by saying that in polycrystals whether a GB switches modes or stagnates does not only depend on the type of GB (and the energy difference for nucleation of different disconnection types) but also on the nature of the constraints (a non-local effect). We also note in the same paragraph that true stagnation probably never happens, because given enough time, we could always nucleated disconnections of other modes. This means that instead of stagnation, we will more likely see very low mobility if other modes are difficult to form. See Lines 291-296. Finally, we note (now in the last paragraph of the “Multiple Disconnection Modes” section), that whether alternative mode disconnections are able to nucleate on any specific GB (for a fixed local environment) will depend on temperature. See Lines 472-475. The bottom line, is the probability for mode switching is NOT a GB property but a property of the GB AND the local environment AND thermodynamic conditions.

Line 255: Should the word "construction" be replaced by "constriction"?

We have rewritten this paragraph to clarify the meaning. See Lines 259-262 in the revised manuscript.

Line 370: Citations should be provided for the known values of (b_i, h_i) for the $\Sigma 13[001](510)$ GB.

For any particular GB, these can be predicted directly from the DSC lattice, as described in A. H. King, D. A. Smith, *Acta Cryst.* A36 (1980) 335-343 and H. Khater, A. Serra, R. Pond, J. Hirth, *Acta Mat.* 60 (2012) 2007–2020. We have added the relevant references and more detail in Lines 440-458.

Line 526 "Discussion": The sentence "The results presented above suggest that shear-coupling is an intrinsic behavior of all grain boundaries under all conditions" is a big assertion. Though it is likely to be true considering the dislocation nature of grain boundaries, I don't think such claim is appropriate before testing it on all types of grain boundary taking into account all the macroscopic and microscopic degrees of freedom. A slight change of tone would be good, like ".....shear-coupling might be an intrinsic.....".

We agree with the Reviewer that this is indeed a major assertion and one which is not proven. However, like the Reviewer, we believe it to be true. However, to be clear, we now label this a “suggestion” and a “speculation” in the first two lines of the Discussion section.

Line 657 "Methods": Does the sentence "....individual grains from spanning the simulation cell....." mean "facilitate individual grains from..." or "prevent individual grains from...."?

Thanks for noting this error. We have rewritten this sentence to correct it. See Lines 687-688.

Line 664 "Methods": $1914 \times 72 \times 18 \text{ \AA}$ should be $1914 \times 72 \times 18 \text{ \AA}^3$. Similar places in the main text and supplementary information should also be changed.

We rewrote the dimensions as “ $1914 \text{ \AA} \times 72 \text{ \AA} \times 18 \text{ \AA}$ ” to avoid any ambiguity.

Moreover, I noticed that there are four videos, which have neither been referred in the main text nor been explained in the supplementary information.

Thanks for noting this oversight. We have added notes referring to these in the captions to Figs. 1, 3 and 4.

Reviewer #3 (Remarks to the Author):

This paper marks a significant milestone in understanding the behavior of grain boundaries in metals - though it is certainly not the end of the journey.

For several decades, research on grain boundaries has proceeded along largely parallel tracks. Grain boundaries in polycrystals are assumed to behave according to simple rules, mostly based on capillarity considerations. Individual grain boundaries, however, are treated in the context of a hierarchy of crystal lattice defects, in which they have individual properties and exhibit behaviors such as linked migration and sliding, which is the focus of much of the present paper.

Until now, the complexity of the linkages between interfaces in a polycrystal has defied attempts to reconcile the two approaches, although various ideas for overcoming some of the challenges have been discussed - mostly over a lot of beer.

This paper makes explicit observations of the effects of individual grain boundary structures in the context of a polycrystal, during grain growth. It goes further, by comparing the behavior of some individual boundaries in cases where they are unconstrained by polycrystalline connectivity with cases where the constraints are crudely modeled. The induction of multiple modes of disconnection-mediated grain boundary motion in constrained cases confirms what had often been postulated zymurgically, but never analyzed in detail up to now. This is a very nice use of computer modeling to find a result that could never be accessed experimentally.

Thanks for the great comments – we appreciate the beer-references! We wonder if this Reviewer would like to be our publicist?

The paper suggests a rationalization for the formation of annealing twins, including some of the observed peculiarities of that phenomenon. This is a long-standing puzzle that has only had very unsatisfactory explanations up to now, more-or-less on the order of "nature abhors an untwined crystal."

Great comment! We'll have to quote this in the future.

Obviously I think this paper is a significant step forward in understanding polycrystalline behavior. There are, however, a few very minor concerns:

First, in terms of nomenclature, I think that in the context of this paper, some fine distinctions are helpful, even if they are not commonly made elsewhere. In particular, the term "anisotropy" (of grain boundary properties) is applied a little loosely. Anisotropy refers to the change in properties of a grain boundary as its plane is rotated (at fixed misorientation.) The variation of energy with misorientation is really NOT anisotropy. In the wonderland of theory, a boundary can have properties that are isotropic AND misorientation-dependent, and I think that the distinction is worth maintaining.

In the case of grain rotation, for example, the two explanations are migration-induced shear (or vice versa) and misorientation-dependent energy, but NOT anisotropy.

Touché. Got it. We removed all offending anisotropies from the document.

On the topic of grain rotation, it is clear, I believe, that all of the observed cases presented in this paper are migration-induced shear. Among the tell-tale signs, the rotation is unidirectional, while it would be expected to be more like a rotational form of Brownian motion for the case of misorientation-dependent energy.

The authors do not imply that grain rotation cannot be driven by misorientation-dependent energy, but perhaps they might wish to make that clear, along with the fact that it is unlikely to be observed over the time and length scales studied in the current work. Indeed, this is the only major challenge with interpreting the results presented here: they relate to competing processes that must certainly have differing temperature, time and length scale dependences, at the individual boundary level as well as at the level of the polycrystalline assembly. The specific results, must therefore be considered with some caution as to their application at different length scales, timescales and temperatures.

Good point. We added a long parenthetical note to address this point in the Section “Simple Polycrystalline Microstructure” – see Lines 195-199. We discuss the time and length scale effects the Reviewer mentions briefly in the Discussion – see Lines 565-570.

That said, the findings of this work are exciting, and the phenomena that have been discovered are likely to be important aspects of understanding polycrystalline behavior going forward. It is, to the best of my knowledge, the first time that grain boundary structure considerations have been applied to polycrystalline behavior in a systematic and logical way.

Agreed (not surprisingly)!

Reviewers' Comments:

Reviewer #2:

Remarks to the Author:

Thank you for your updated manuscript. It is now suitable for publication.

Reviewer #3:

Remarks to the Author:

The authors have fully addressed the concerns in the prior review.

Response to Reviewers

Reviewer #2 (Remarks to the Author):

Thank you for your updated manuscript. It is now suitable for publication.

We thank the Reviewer for their assistance improving this article.

Reviewer #3 (Remarks to the Author):

The authors have fully addressed the concerns in the prior review.

We thank the Reviewer for their assistance improving this article.

There were additional changes made in response to the editorial requests. Figure labels and captions, as well as subject headings were updated to be consistent with the *Nature* style guidelines. The supplementary notes were also reformatted to be consistent with *Nature* guidelines. A typographical error was corrected in Equation 12 of the Supplementary Notes. A brief discussion of conclusions and broader impact was added to the end of the abstract as well as the final paragraph in the Introduction section.